# A Novel Multi-Target Small Molecule, LCC-09, Inhibits Stemness and Therapy-Resistant Phenotypes of Glioblastoma Cells by Increasing miR-34a and Deregulating the DRD4/Akt/mTOR Signaling Axis

**DOI:** 10.3390/cancers11101442

**Published:** 2019-09-26

**Authors:** Ya-Ting Wen, Alexander TH Wu, Oluwaseun Adebayo Bamodu, Li Wei, Chien-Min Lin, Yun Yen, Tsu-Yi Chao, Debabrata Mukhopadhyay, Michael Hsiao, Hsu-Shan Huang

**Affiliations:** 1The PhD Program for Translational Medicine, College of Medical Science and Technology, Taipei Medical University and Academia Sinica, Taipei 110, Taiwan; 98142@w.tmu.edu.tw (Y.-T.W.); chaw1211@tmu.edu.tw (A.T.W.); 2Department of Neurosurgery, Taipei Medical University-Wan Fang Hospital, Taipei 110, Taiwan; 3Graduate Institute of Medical Sciences, National Defense Medical Center, Taipei 114, Taiwan; 4Department of Hematology and Oncology, Cancer Center, Taipei Medical University—Shuang Ho Hospital, New Taipei City 235, Taiwan; 16625@s.tmu.edu.tw (O.A.B.); 10565@s.tmu.edu.tw (T.-Y.C.); 5Department of Medical Research & Education, Taipei Medical University—Shuang Ho Hospital, New Taipei City 235, Taiwan; 6Graduate Institute of Injury Prevention and Control, College of Public Health, Taipei Medical University, Taipei 110, Taiwan; nsweili@gmail.com; 7Division of Neurosurgery, Department of Surgery, Taipei Tzu Chi Hospital, Buddhist Tzu Chi Medical Foundation, New Taipei City 231, Taiwan; 8Department of Neurology, School of Medicine, College of Medicine, Taipei Medical University, Taipei 110, Taiwan; m513092004@tmu.edu.tw; 9Division of Neurosurgery, Department of Surgery, Taipei Medical University—Shuang Ho Hospital, New Taipei City 235, Taiwan; 10Taipei Neuroscience Institute, Taipei Medical University, Taipei 110, Taiwan; 11Graduate Institute for Cancer Biology and Drug Discovery, College of Medical Science and Technology, Taipei Medical University, Taipei 110, Taiwan; yyen@tmu.edu.tw; 12TMU Research Center of Cancer Translational Medicine, Taipei Medical University, Taipei 110, Taiwan; 13Taipei Cancer Center, Taipei Medical University, Taipei 110, Taiwan; 14Department of Biochemistry and Molecular Biology, Mayo Clinic, Rochester, MN 55905, USA; Mukhopadhyay.Debabrata@mayo.edu; 15Genomics Research Center, Academia Sinica, Taipei City 11529, Taiwan; 16Department of Biochemistry, Kaohsiung Medical University, Kaohsiung 800, Taiwan

**Keywords:** LCC-09, glioblastoma (GBM), glioma stem cells, stemness, temozolomide resistance, dopamine receptor, miR-34a, maintenance therapy, adjuvant therapy, multi-target therapeutics

## Abstract

The management of glioblastomas (GBMs) is challenged by the development of therapeutic resistance and early disease recurrence, despite multi-modal therapy. This may be attributed to the presence of glioma stem cells (GSCs) which are known to survive radio- and chemotherapy, by circumventing death signals and inducing cell re-population. Recent findings suggest GSCs may be enriched by certain treatment modality. These necessitate the development of novel therapeutics capable of targeting GBM cell plasticity and therapy-resistant GSCs. Here, aided by computer-assisted structure characterization and target identification, we predicted that a novel 5-(2′,4′-difluorophenyl)-salicylanilide derivative, LCC-09, could target dopamine receptors and oncogenic markers implicated in GBMs. Bioinformatics data have indicated that dopamine receptor (DRD) 2, DRD4, CD133 and Nestin were elevated in GBM clinical samples and correlated to TMZ (Temozolomide) resistance and increased ALDH (Aldehyde dehydrogenase) activity (3.5–8.9%) as well as enhanced (2.1–2.4-fold) neurosphere formation efficiency in U87MG and D54MG GBM cell lines. In addition, TMZ-resistant GSC phenotype was associated with up-regulated DRD4, Akt, mTOR, β-catenin, CDK6, NF-κB and Erk1/2 expression. LCC-09 alone, or combined with TMZ, suppressed the tumorigenic and stemness traits of TMZ-resistant GBM cells while concomitantly down-regulating DRD4, Akt, mTOR, β-catenin, Erk1/2, NF-κB, and CDK6 expression. Notably, LCC-09-mediated anti-GBM/GSC activities were associated with the re-expression of tumor suppressor miR-34a and reversal of TMZ-resistance, in vitro and in vivo. Collectively, these data lay the foundation for further exploration of the clinical feasibility of administering LCC-09 as single-agent or combinatorial therapy for patients with TMZ-resistant GBMs.

## 1. Introduction

Glioblastoma (GBM), a WHO grade IV astrocytoma which accounts for about 45% of all malignant tumors of the primary central nervous system (CNS) and 54% of all gliomas, is one of the most fatal, frequently diagnosed, and highly aggressive brain malignancies in adults, with an average annual incidence of 3–4 in 100,000 and median survival of approximately 1.25 years [1,2]. Despite comprehensive molecular characterization, increasing understanding of its biology, and touted improvement in diagnostic strategies and treatment modality, GBM remains a therapeutic enigma being characterized by enhanced propensity to relapse, early recurrence, innate non-responsiveness to therapy or acquisition of resistance to initially effective anticancer therapy, and high incidence of disease-specific death, usually within the first year of diagnosis or treatment initiation [1,3,4]. This high capacity for dissemination and therapeutic intractability of GBM may not be unassociated with disease-specific complex inter- and intra-tumoral heterogeneity, which is driven by the presence and activities of a small subset of glioma cells with intrinsic ability to form heterogeneous glial tumors and characterized by enhanced tumorigenicity, multi-potency and unrestrained self-propagation; these cells are herein referred to as glioma stem cells (GSCs) [5,6]. The ability of the GSCs to initiate and sustain GBM, as well as resist current conventional anti-GBM therapies, makes them important therapeutic targets and necessitates the development of novel therapeutic approaches with demonstrable high curative efficacy. The transcription factor nuclear factor kappa-B (NF-κB) is increasingly implicated in the maintenance and propagation of tumor-initiating cells (TICs) and induction of associated cancer stem cell (CSC)-like phenotypes [2,7]. In the last decade, accumulating evidence indicate that NF-κB signaling plays a critical role in mesenchymal differentiation and the propagation of GSCs [7,8,9,10]. The NF-κB signaling has been shown to facilitate oncogenic transformation, maintain the aggressiveness of cancerous cells, and more importantly, enhance the tumor microenvironment (TME) by sustaining and/or expanding the GSC population [10,11,12].

The above is contextually relevant, especially as aberrant expression of the receptor activator of NF-κB ligand (RANKL) characterizes highly metastatic glioma cells, and the constitutively high expression of RANKL in GBM cells induces the activation of neighboring astrocytic glial cells in the TME through NF-κB signaling [9,13]. In a Nature Review publication, Linda Koch suggested the inhibition of RANKL as a probable new anticancer strategy [14], and this has been re-echoed even more recently, as the inhibition of RANKL/RANK signaling, which regulates progesterone-mediated development of the lactating mammary gland, breast stem cell proliferation and expansion, and drives the formation of hormone-induced breast cancer, is again flaunted as a likely effective anticancer therapeutic option [15]. Therapeutically-relevant, our team recently synthesized a series of new 5-(2′,4′-difluorophenyl)-salicylanilide derivatives, and demonstrated that these new class of salicylanilides effectively inhibit the expression and/or activities of RANKL and also suppress RANKL-related effector genes, including NF-κB, nuclear factor of activated T cells 1 (NFATc1), c-fos, triiodothyronine receptor auxiliary protein (TRAP), and cathepsin K [16]. Thus, this proof-of-concept study tested the therapeutic efficacy of one of our new 5-(2′,4′-difluorophenyl)-salicylanilide derivatives, with the International Union of Pure and Applied Chemistry (IUPAC) name N-(3-cyanophenyl)-2′,4′-difluoro-4-hydroxy[1,1′-biphenyl]-3-carboxamide, which is herein referred to as LCC-09, and consists of the functional fragment of magnolol, 2,4-difluorophenyl, and paeonol moiety of salicylate. Interestingly, LCC-09 not only demonstrated significant suppression of RANKL- and, by inference, NF-κB-mediated GSC phenotypes, but also elicited a very strong inhibitory effect on the expression levels of dopamine receptors 2 (DRD2), and 4 (DRD4) proteins, CDK6, β-catenin, and Akt/mTOR, all of which are implicated in GBM mitogenic signaling and expansion of the GSC [17,18]. Thus, the present study highlights the putative role of LCC-09 as a novel small molecule multi-target antagonist with significant anti-GSC therapeutic efficacy.

## 2. Materials and Methods

### 2.1. Drugs and Chemicals

Temozolomide (TMZ; ≥98% HPLC, T2577) was purchased from Sigma-Aldrich, while LCC-09 was synthesized as previously described (16). Stock solution of TMZ or LCC-09 were dissolved in dimethyl sulfoxide (10 mg/mL) (DMSO; Sigma Aldrich Co., St. Louis, MO, USA) and stored at −20 °C away from light before use, then diluted in sterile culture medium just before use. Gibco^®^ RPMI-1640, fetal bovine serum (FBS), trypsin/EDTA, phosphate buffered saline (PBS), sulforhodamine B (SRB) medium, tris(hydroxymethyl)aminomethane (TRIS) base and acetic acid were all purchased from Sigma Aldrich Co. (St. Louis, MO, USA), A BCA protein assay was purchased from Pierce (Rockford, IL, USA) and the polyvinylidene difluoride (PVDF) membrane was purchased from Millipore (Bedford, MA, USA).

### 2.2. Cell Lines and Culture

The human GBM cell lines, U87MG and D54MG, obtained from American Type Culture Collection (ATCC. Manassas, VA., USA) were cultured in RPMI-1640, supplemented with 10% FBS and 1% penicillin/streptomycin (Invitrogen, Life Technologies, Carlsbad, CA, USA) and incubated at 37 °C in 5% humidified CO_2_ incubator. The cells were sub-cultured at 96–98% confluence or the culture media changed every 72 h. After initial dose–response evaluation (100 to 1000 μM), TMZ-resistant U87MG and D54MG cells were established by continuous exposure to the IC_50_ of TMZ for ~6 months. For the colorimetric drug cytotoxicity (cell viability) assays, the cells were treated with specified concentrations of TMZ and/or LCC-09 over different duration of time.

## 3. Western Blot Analysis

10 μg protein lysates derived from U87MG and D54MG cells which were incubated with or without indicated concentrations of LCC-09 and/or TMZ over specified time durations, were ran on 10% SDS-PAGE gel and then transferred to PVDF membranes using the Trans-Blot^®^ Turbo^TM^ transfer system (Bio-Rad Laboratories, Inc., Hercules, CA, USA). We blocked non-specific binding by incubating the membranes with 5% non-fat milk in Tris-buffered saline (TBS) with Tween 20 (TBST) for 1 h, followed by incubation at 4 °C overnight with antibodies against β-catenin (1:1000, #8814, Cell Signaling Technology, Danvers, MA, USA), DRD4 (1:1000, sc-136169, Santa Cruz Biotechnology, Inc., Santa Cruz, CA, USA), mTOR (1:1000, #2983, Cell Signaling Technology, Danvers, MA, USA), Akt (1:1000, #4691, Cell Signaling Technology, Danvers, MA, USA), NF-κB p65 (1:1000, #8242, Cell Signaling Technology, Danvers, MA, USA), phospho-p44/42 MAPK (Erk1/2) (1:2000, #4370, Cell Signaling Technology, Danvers, MA, USA), p44/42 MAPK (Erk1/2) (1:1000, #4695, Cell Signaling Technology, Danvers, MA, USA), CDK6 (1:1000, #3136, Cell Signaling Technology, Danvers, MA, USA), Stat3 (1:1000, sc-482, Santa Cruz Biotechnology, Inc., Santa Cruz, CA, USA), c-Myc (1:1000, sc-24580, Santa Cruz Biotechnology, Inc., Santa Cruz, CA, USA), and β-actin (1:500, sc-58673, Santa Cruz Biotechnology, Inc., Santa Cruz, CA, USA). After incubation with primary antibodies overnight, the membranes were then incubated with horseradish peroxidise (HRP)-linked secondary antibodies for 1 h at room temperature (RT) and washed thrice with 1× PBS. The protein bands were detected using the enhanced chemiluminescence (ECL) detection system (Thermo Fisher Scientific Inc, Waltham, MA, USA)

### 3.1. miR-34a Transfection

For miR-34a transfection, when 3.5 × 10^4^ GBM cells seeded per well in 6-well plates attained ~70–80% confluence, depending on the experimental group, they were transfected with either miR-34a mimic, miR-34a inhibitor, or miR-34a negative control (Shanghai GeneChem, Inc., Shanghai, China) using Invitrogen^®^ Lipofectamine^TM^ 2000 transfection reagent (#11668019, Thermo Fisher Scientific Inc, Waltham, MA, USA) according to manufacturer’s instruction. The total RNA or protein were extracted 48 h after transfection and used for the subsequent cellular or biomolecular analyses.

### 3.2. RNA Extraction and RT-PCR

After total RNA extraction from GBM cells using RNeasy kit (Qiagen, Inc., Gaithersburg, MD, USA) and miRNA purification using miRNEasy kit (Qiagen), RNA concentration and purity was determined using the NanoDrop 1000 spectrophotometer (Nyxor Biotech, Paris, France). RNA was reverse-transcribed into cDNA using the Ultrapure SMART^®^ recombinant Moloney Murine Leukemia Virus (MMLV) Reverse Transcriptase (#639523, Takara Bio USA, Inc., Shiga, Japan) following the manufacturer’s instructions. PCR was performed with SYBR-Green Master Mix (Applied Biosystems, Life Technologies, Grand Island, NY, USA). Amplification reactions were performed using the following conditions: 95 °C for 3 min, 35 cycles at 95 °C for 15 s, 60 °C for 30 s, 72 °C for 30 s, and 72 °C for 10 min. The 2^−ΔΔCT^ method was used for quantification of the mRNA expression of miRs. β-actin served as internal control and was used for normalization of all mRNA values. The PCR primers for hsa-miR-22_1 (#MS00003220), hsa-miR-26b_1 (#MS00003234), hsa-miR-34a*_1 (#MS00009534), hsa-miR-98_1 (#MS00003367), hsa-miR-98-3p_1 (#MS00045136), hsa-miR-142-3p_2 (#MS00031451), hsa-miR-143*_1 (MS00008687), and hsa-miR-145*_1 (#MS00008708) were all purchased from QIAGEN (QIAGEN Inc., Germantown, MD, USA).

### 3.3. Sulforhodamine B (SRB) Cell Viability Assay

U87MG or D54MG cells (4 × 10^3^) were seeded per well in 96-well plates and incubated for 24 h, thereafter the cells were treated with different concentrations of LCC-09 and/or TMZ. After treatment for 48 h, the treated GBM cells were washed in 1× PBS thrice. They were then fixed using 10% Trichloroacetic acid (TCA) for 1 h, and washed with ddH_2_O two times, before the viable cells were incubated in 0.4% SRB (*w/v*)/1% acetic acid solution at RT for 1 h. The unattached SRB dye was removed by carefully washing with 1% acetic acid thrice and then the plates were air-dried. The bound dye was dissolved in 10mM Trizma base, and absorbance was read at a wavelength of 570 nm in the Synergy Neo2 Hybrid multi-mode microplate reader system (BioTek Instruments, Inc., Winooski, VT, USA).

### 3.4. Combination Median Effect Analysis

For quantitative assessment of the nature of LCC-09 and TMZ interaction, we used the Chou–Talalay algorithm-based isobologram method. The combination of TMZ with LCC-09 was in fixed ratio of indicated concentrations. The CompuSyn software (ComboSyn, Inc., Paramus, NJ, USA) was used for combination index (CI) estimation. CI = 1, CI <1 or CI >1 were interpreted as additive, synergistic, or antagonistic, respectively. Similarly isobolograms were also drawn based on the concentrations of LCC-09 and/or TMZ that induced 50%, 75%, and 90% inhibition of viability; the drug combination data points that fall on the hypotenuse, within the right-angled isobologram triangle, or outside the right-angled isobologram triangle were interpreted as additive, synergistic, or antagonistic effect, respectively.

### 3.5. Neurosphere Formation Assay

For neurosphere generation, we cultured parental or CD133^hi(+)^ TMZ-resistant U87MG and D54MG cells in HEScGRO^TM^ serum-free medium for human embryonic stem cell culture (SCM020, Merck KGaA, Darmstadt, Germany), supplemented with 20 ng/mL hEGF (Millipore, Bedford, MA), 10 ng/mL hbFGF (Invitrogen, Carlsbad, CA), heparin (#07980; STEMCELL Technologies Inc., Interlab Co., Ltd, Taipei, Taiwan), B27 supplement (Invitrogen, Carlsbad, CA), and NeuroCult^TM^ NS-A proliferation supplement ( Human; #05753; STEMCELL Technologies Inc. Interlab Co., Ltd, Taipei, Taiwan). GBM cells were seeded (1000 cells per mL/well) in 6-well ultra-low adhesion plates (Corning Inc., Corning, NY) and cultured for 10–12 days. The anchorage-independent neurospheres [diameter (⌀) ≥ 100 µm] were counted, and photographed using inverted phase-contrast microscope.

### 3.6. Colony Formation Assay

2 × 10^4^ U87MG or D54MG cells pre-treated with or without indicated concentration of LCC-09 for 48 h were seeded per well in triplicate in 6-well culture plates, and incubated in a 5% CO_2_ atmosphere incubator at 37 °C for 12 days. Formed colonies (⌀ ≥100 µm) made up of ≥50 cells, were washed with PBS twice, fixed with cold methanol for 15 min, stained with 0.5% crystal violet for 15 min at RT, then observed and counted. The number and size of colonies formed were estimated with the ChemiDoc-XRS imager (QuantityOne software package; Bio-Rad, Hercules, CA, USA).

### 3.7. Scratch Wound-Healing Migration Assay

U87MG or D54MG (3 × 10^5^) treated with or without LCC-09, were seeded in 6-well plates and incubated in 5% CO_2_ atmosphere incubator for 48 h to attain 100% confluence. Equal-sized scratch wounds were made along the medial axes of the plates containing confluent adherent cells with the aid of sterile 200 µL micropipette tips. Detached cells were removed by carefully washing with 1× PBS thrice, then the adherent cells were incubated in new culture media at 37  °C in 5% CO_2_ humidified incubator to allow wound closure/healing. Closure of the denuded area was monitored and photographed at the indicated time-points under microscope with a 10× objective lens, and analyzed with the NIH ImageJ software (https://imagej.nih.gov/ij/).

### 3.8. Fluorescence-Activated Cell Sorting (FACS)

Parental or TMZ-resistant U87MG or D54MG cells were detached from culture plates using trypsin-EDTA and washed with PBS solution containing 0.1% BSA. Then 1 × 10^6^ cells in 100 µL PBS solution containing 0.5% BSA were incubated with allophycocyanin (APC)-conjugated fluorescence-labeled mouse anti-human CD133/2 (Miltenyi Biotec, Bergisch Gladbach, Germany) or APC-conjugated isotype control mouse IgG_2b_ in the dark for 15 min at RT. The isotype control was used for appropriate gating. The labeled U87MG or D54MG cells were sorted by flow-cytometry into CD133+ or CD133− groups, and the data collected and analyzed using the BD FACSCanto^TM^ flow cytometry system (BD Biosciences, CA, USA).

### 3.9. ALDEFLUOR ALDH Activity Analysis

For determination of aldehyde dehydrogenase (ALDH) activity analysis, we used the ALDEFLUOR^TM^ Kit (STEMCELL Technologies, Cambridge, UK) according to the manufacturer’s instruction. Briefly, 1 × 10^6^ U87MG or D54MG cells cultured with or without 10 μM rapamycin, were re-suspended in 1 mL ALDEFLUOR buffer. After washing the U87MG or D54MG cells in ALDEFLUOR buffer, they were maintained at 4 °C throughout the cell staining process. ALDH activity was assessed and cells differentiated into ALDH-high (ALDH^hi^) or ALDH-low (ALDH^low^) using the fluorescence (FL1) and low side scatter (SCC) channels of a BD FACSCanto^TM^ flow cytometry system (BD Biosciences, CA, USA) and FACSDiva software (v 6.1.2, BD Biosciences, CA, USA). Cells were sorted with FACS based on fluorescence intensity corresponding to their ALDH activity levels. ALDH^hi^ and ALDH^low^ cells were harvested and cultured separately.

### 3.10. Animal Studies

The in vivo study was approved by and performed in accordance to the Affidavit of Approval of Animal Use Protocol, Taipei Medical University (Approval No. LAC-2017-0161). 6-weeks old female NOD/SCID mice (*n* = 20, median weight 25.69 ± 1.83 g) were purchased from BioLASCO Taiwan Co., Ltd (Taipei, Taiwan) and maintained under specific-pathogen-free conditions in the Laboratory Animal Center (LAC), Taipei Medical University,. The mice were randomly placed into either of two experimental models. In the subcutaneous tumor model, mice (*n* = 8) were subcutaneously inoculated in the right hind flank with 5 × 10^5^ luciferase-expressing parental or TMZ-resistant U87MG cells suspended in 0.1 mL PBS. In the orthotopic tumor model, TMZ-resistant U87MG neurospheres (1 × 10^5^ cells/injection) were orthotopically injected into NOD/SCID mice (*n* = 12) using previously established protocol [19]. Tumor growth in both experimental models was monitored using Xenogen™ IVIS 200 bioluminescence imaging system (#IVIS200-BR-01; Caliper Life Science Inc., Hopkinton, MA, USA) once every week. The change in tumor burden was determined by total photon flux (photons/s), which was calculated as flux per unit area and unit angle over the region of interest (ROI). On day 8 post-tumor inoculation, the subcutaneous or orthotopic experimental mice models were again randomly placed into vehicle-treated control (*n* = 5), LCC-09 (*n* = 5), TMZ (*n* = 5), or LCC-09 + TMZ combination (*n* = 5) treatment groups. Mice in the vehicle group were treated with 0.5 mL PBS; LCC-09 treatment group were injected intraperitoneally (i.p.) with 5 mg/kg LCC-09, 5 times/week; TMZ treatment group received 42 mg/kg TMZ by oral gavage, 5 times/week, according to a previously published protocol [20]; and the combination group received 5 mg/kg LCC-09 + 42 mg/kg TMZ as earlier described. The mice body weights were recorded once every week. Mice survival was monitored over the duration of the experiment. The mice alive after the experiments were humanly sacrificed and the tumor samples were collected for further analyses.

### 3.11. Statistical Analysis

Data presented represents the means ± SD of experiments performed at least 4 times in triplicates. Comparison between two groups was estimated using the 2-sided Student’s *t*-test, while the one-way analysis of variance (ANOVA) was used for comparison between 3 or more groups. *P*-value < 0.05 was considered statistically significant. All statistical analyses were carried out using the IBM SPSS statistics for Windows, version 25.0 (IBM Corp. Released 2016. Armonk, NY, USA).

## 4. Results

### 4.1. The Design, Synthesis and Functional Characterization of the Novel Multi-Target Small Molecule Dopamine Antagonist, Lcc-09

Using a unique approach for discovery of new chemical entities; we developed several novel series of 5-(2′,4′-difluorophenyl)-salicylanilide derivatives based on difluorobiphenyl and salicylanilide scaffolds (Figure 1A); one of which is the LCC-09, consisting of the functional fragments of magnolol, 2,4-difluorophenyl, and paeonol moiety of salicylate (Figure 1B, see also Appendix A). Computer-assisted structure characterization and target prediction partly based on the broadly-used “scaffold identification” and “ligand-based” chemical fingerprint strategy was used to identify proteins with known ligands similar to our query molecule [21], and predict LCC-09 targets dopamine receptors DRD1, DRD2, DRD3, and DRD4 (Figure 1C), as well as oncogenic markers implicated in GBM metastatic and recurrent phenotype including EGFR, Akt, mTOR, Erk1/2, NF-κB, c-Myc, β-catenin, CDK6, and EZH2 (Figure 1D). Since the ability of small molecules inhibitors to cross the blood-brain-barrier (BBB) is essential for their therapeutic efficacy in preventing or treating brain tumors and/or metastasis [22], using bioinformatics-aided prediction of LCC-09 adsorption, distribution, metabolism and excretion (ADME), druglikeness and medicinal friendliness, we demonstrated that the 350 g/mol LCC-09 exhibits a rate of brain penetration of −1.3, extent of brain penetration of −0.01, and brain/plasma equilibration rate of −3 (Appendix A).

### 4.2. Enhanced cd133 Positivity and Constitutive drd4 Dopaminergic Signaling Characterize TMZ Resistance in gbm Cells

Based on the results of our preliminary computer-assisted chemical fingerprinting for the structural and functional characterization, which suggested the pharmacological targetability of DRDs by LCC-09, we evaluated the expression profile of DRDs in the clinical samples from patients with high grade GBM or low grade glioma (LGG) by accessing and statistically analyzing the Gene Expression Omnibus (GEO) GSE8692/GPL96 dataset (*n* = 12, 22,283 probes) (https://www.ncbi.nlm.nih.gov/gds/?term=GSE8692). Our results showed that while no difference was observed between the expression of DRD1 in the high grade GBM and LGG, the GBM group had higher expression level of DRD2, DRD3, and DRD4 mRNA compared to the LGG group, and this expression profile positively correlated with that of GSC markers, Nestin, and CD133 (Figure 2A). Against the background that CD133 and Nestin are broadly implicated in the resistance of GBM cells to TMZ treatment [23,24], using the established TMZ-resistant U87MG and D54MG cells, we showed that relative to their parental counterparts, CD133 positivity was enhanced 4.02- and 6.35-fold in the TMZ-resistant U87MG and D54MG cells, respectively (Figure 2B). We also demonstrated that the observed increased CD133 positivity in the TMZ-resistant cells was positively correlated with concurrent enhanced expression of DRD2 (U87MG: *p* < 0.01; D54MG: *p* < 0.001) and DRD4 (U87MG: *p* < 0.01; D54MG: *p* < 0.01) (Figure 2C). In addition, the CD133^+^ TMZ-resistant cells generated a significantly higher number of neurospheres, compared to their parental U87MG (~38% increase, *p* < 0.01) and D54MG (~27% increase, *p* < 0.01) cells (Figure 2D). Furthermore, in a bid to delineate the association between TMZ-resistance, CD133 positivity, and DRD4 expression, using the xenograft murine GBM models, we demonstrated that compared to the 33.3% (*n* = 3) tumorigenicity observed in litter-mates inoculated with parental U87MG cells, all the mice (*n* = 3) inoculated with CD133^+^ TMZ-resistant cells recorded tumor formation by day 21 (Figure 2E). More interestingly, using Western blot analyses, we showed that this enhanced tumorigenicity was associated with concurrently upregulated expression of DRD4, β-catenin, Akt, mTOR, NF-κB, Erk1/2, p-Erk1/2, and CDK6 (Figure 2F, see also Appendix A). These data, at least in part, indicate the existence of a positive correlation between CD133 positivity and DRD4 expression, as well as a probable functional association between the duo and resistance to TMZ in GBM.

### 4.3. LCC-09 Suppressed the Viability and Metastatic Phenotype of GBM Cells through the Dysregulation of DRD4-Mediated AKT/mTOR and NF-κB Signaling Axes

Having established the role of DRD4 in the tumorigenicity, metastatic and GSC-like phenotypes of GBM, as well as its association with CD133 positivity, we then investigated the probable inhibitory effect and therapeutic efficacy of LCC-09 on the CD133^+^ TMZ-resistant U87MG or D54MG cells in particular. Using the SRB colorimetric assay, we showed that low dose LCC-09 significantly inhibited the viability of the CD133^+^ TMZ-resistant U87MG or D54MG cells in a dose-dependent manner (Figure 3A). With IC_50_ of 3.6 μM and 5.8 μM for the CD133^+^ TMZ-resistant U87MG or D54MG cells, respectively, we noted that 48 h exposure to 8 μM LCC-09 elicited a ~75% and ~61% reduction in cell viability of the CD133^+^ TMZ-resistant U87MG or D54MG cells, respectively (Figure 3A). Since the ability to form colonies in distant anatomic sites is crucial for GBM progression and recurrence, we tested the effect of LCC-09 on biologic property of GBM cells. Of translational relevance, we demonstrated that treatment with only 3 μM LCC-09 markedly suppressed the clonogenicity of the CD133^+^ TMZ-resistant U87MG (1.69-fold, *p* < 0.01) or D54MG (2.47-fold, *p* < 0.001) cells (Figure 3B). Furthermore, using the scratch-wound healing migration assay, we demonstrated that 8 h exposure to 3 μM LCC-09 significantly attenuate the motility of TMZ-resistant cells, as expressed by a 48.7% (*p* < 0.01) or 8.8% (*p* < 0.05) lag in the motility of CD133^+^ TMZ-resistant U87MG or D54MG cells, respectively, in comparison to their untreated counterpart (Figure 3C). To better understand the underlying mechanism for the LCC-09-mediated reduction in GBM cell viability, clonogenicity, and migration, using Western blot assays, we evaluated the expression of our selected panel of oncogenic proteins, and observed that 3 μM LCC-09 significantly reduced the expression level of DRD4 protein, and this was associated with the concurrent downregulation of β-catenin, Akt, mTOR, Erk1/2, p-Erk1/2, NF-κB, c-Myc, and CDK6 protein expression levels (Figure 3D, see also Appendix A). These results are indicative of the ability of LCC-09 to inhibit the viability, migration, and clonogenicity of CD133^+^ TMZ-resistant U87MG and D54MG cells, partly by suppressing DRD4 expressing and consequently dysregulating the DRD4-mediated AKT/mTOR and NF-κB signaling axes.

### 4.4. LCC-09, by Targeting Dopaminergic Signals and ALDH Activity, Induces Marked Attenuation of the Stem Cell-Like Phenotype of GBM Cells

GSCs are characterized by enhanced proliferative, metastatic, and drug-resistant traits, as well as high clonogenicity and neurosphere formation efficiency compared to other cells in the tumor niche [25,26,27]; we further evaluated the effect of LCC-09 on these GSC-like phenotypes of the CD133^+^ TMZ-resistant U87MG or D54MG cells. Using the neurosphere formation assay, we demonstrated that the treatment of these TMZ-resistant cells with 2–10 μM LCC-09 for 24 h inhibited the formation of neurospheres, dose-dependently, with 24 h exposure to 8 μM eliciting a 73% (*p* < 0.001) or 49% (*p* < 0.001) reduction in number of neurospheres formed by the U87MG or D54MG cells, compared to their vehicle-treated counterparts (Figure 4A). Understanding that ALDH activity, which is a marker of cancer stemness, is also implicated in metastasis, therapy failure, and poor prognosis [28], we demonstrated that compared to the parental or vehicle-treated neurospheres, 6 μM LCC-09 suppressed ALDH activity in the D54MG neurospheres by 34.4% or 59.1%, respectively; while in the U87MG neurospheres, it elicited a 30.2% or 42.1% reduction, respectively (Figure 4B). Consistent with earlier results, the observed LCC-09-mediated suppression of neurosphere formation efficiency and ALDH activity was associated with concurrent downregulation of DRD4, β-catenin, Akt, mTOR, Erk1/2, p-Erk1/2, EZH2, c-Myc, and CDK6 protein expression levels in the ALDH^hi^ neurospheres (Figure 4C, also see Appendix A). These data indicate that by targeting dopaminergic signals and ALDH activity, LCC-09 effectively induces attenuation of the stem cell-like phenotype of GBM cells.

### 4.5. LCC-09 Synergistically Enhances the Anticancer Effect of TMZ in Therapy-Resistant GBM Cells

Since TMZ-resistance is the Achilles’ heel of the therapy of choice for patients with GBM [3,4,5], and having shown that LCC-09 exhibits significant anti-GSC effects, we thus examined if LCC-09 could be combined with TMZ, and the effect of such combination on the anticancer activity of the latter. Results of our cytotoxicity assay show that treatment with 1–2 μM LCC-09 alone or combined with 200–500 μM TMZ sequentially, reduced the viability of the D54MG or U87MG cells, and significantly enhanced the anticancer effect of TMZ on the cells in a dose-dependent manner (Figure 5A). Using the Chou–Talalay-based algorithm for drug combination analyses, we observed that all drug combination points were located within the right-angle isobologram triangle and that all CI values were <1, thus, demonstrating that the combined effect of LCC-09 and TMZ against the D54MG or U87MG cells was synergistic (Figure 5B). Compared with the expression levels when treated with TMZ or LCC-09 alone, TMZ/LCC-09 combination therapy significantly downregulated the expression level of DRD4, Akt, mTOR, Erk1/2, p-Erk1/2, CDK4, and CDK6 (Figure 5C, also see Appendix A). These data indicate that LCC-09 synergistically enhances the anticancer efficacy of TMZ in GBM cells.

### 4.6. LCC-09 Suppresses GSC-Related Oncogenic and Dopaminergic Signals While Upregulating miR-34a in GBM Cells, In Vitro

The last decade has been characterized by increasing documentation of the critical role of microRNAs (miRNAs, miRs) in the regulation of GSC activity, including disease progression, therapy failure, and poor prognosis [29,30]. Thus, we evaluated the effect of 24 h exposure of CD133^+^ALDH^hi^DRD4^+^ TMZ-resistant D54MG or U87MG cells to 6 μM LCC-09 on a small panel of carefully curated miRs. Results of our RT-PCR demonstrated that only miR-34a and miR-143/145 cluster mRNA levels were significantly elevated in both the U87MG and D54MG cells (Figure 6A), suggesting, in part, a potential common regulatory mechanism. Since miR-34a suppresses the proliferation and growth of GSCs by targeting multiple SC-associated oncogenic signaling cascades [30,31] including those relevant to the present study as illustrated (Figure 6B), to better understand this LCC-09-induced epigenetic modulation of GBM cells we examined the effect of altered miR-34a expression on GBM cells using miR-34a mimic or inhibitor. Our results indicate that transfection with miR-34a mimic resulted in the suppression of mTOR, DRD4, Erk1/2, CDK6, and EZH2 in the D54MG or U87MG cells. Conversely, transfection with miR-34a inhibitor enhanced the expression of mTOR, DRD4, Erk1/2, CDK6, and EZH2 proteins (Figure 6C). In similar experiments performed under serum-deprived conditions, we observed that GBM cells transfected with miR-34a mimic significantly lost the ability to form neurospheres while those bearing the miR-34a inhibitor exhibited enhanced ability to form neurospheres (Figure 6D).

### 4.7. LCC-09 Significantly Suppresses GSC-Related Oncogenic and Dopaminergic Signals While Up-Regulating miR-34a in GBM Cells, In Vivo

Having shown that LCC-09 suppresses GSC-related oncogenic and dopaminergic signals by upregulating miR-34a in GBM cells, in vitro, we further sought in vivo validation of these in vitro findings; thus, using xenograft murine GBM models established by inoculation with TMZ-resistant U87MG cells, we evaluated the effect of 5 mg/mL LCC-09 administered intraperitoneally on the tumorigenicity and GSC phenotype of GBM cells. Mice were randomly divided into vehicle-treated control (*n* = 5), TMZ-treated (*n* = 5), LCC-09 treated (*n* = 5), and TMZ+LCC-09 -treated (*n* = 5) groups. We demonstrated that while the tumors grew exponentially in the control group, the tumor burden in the LCC-09-treated mice was markedly reduced. Our bioluminescence imaging (BLI) for tumor growth monitoring showed that while the differential tumor-growth in the vehicle-treated control and TMZ-treated mice was unapparent over the course of the experiment, except for week 4 post-tumor inoculation with a 1.63-fold lag in tumor-growth in TMZ mice compared to control mice, LCC-09 treatment, compared to control, reduced the tumor burden by 1.78-fold, and 4.71-fold (*p* < 0.001) at weeks 4 and 5 post-tumor inoculation; interestingly, compared to the control and TMZ-treated groups, concomitant exposure to TMZ and LCC-09 elicited ~200-fold reduction (*p* < 0.001) in tumor-growth by week 5 (Figure 7A,B). In addition, compared to the 40% mortality rate of the control and TMZ-treated mice, the LCC-09-treated and TMZ+LCC-09-treated group elicited 80% and 100% survival rates, respectively (Figure 7C). Moreover, no case of dyscrasia or significant change in body weight was observed between the mice in the 4 treatment groups during the study (Figure 7D). In parallel assays using xenograft-derived cells, we demonstrated that cells from the LCC-09-treated and LCC-09+TMZ combination groups profoundly lost their capacity to form neurospheres quantitatively and qualitatively compared to the control group, as demonstrated by a ~4-fold (*p* < 0.001) reduction in the number of neurospheres formed, in comparison to the vehicle-treated control or TMZ-treated group (Figure 7E). This LCC-09-impaired neurosphere formation efficiency was strongly associated with co-suppressed DRD4, AKT, CDK6, EZH2, and BCL-2 proteins expression level (Figure 7F), with converse significantly upregulated expression of miR-34a, compared to the vehicle-treated group (4.42-fold, *p* = 0.004; Figure 7G) Concomitantly, treatment with LCC-09 alone or combined with TMZ enhanced the expression of miR-34a by ~4.47-fold (*p* = 0.001) or 3.61-fold (*p* = 0.005) compared to the single-agent TMZ-treated group (Figure 7G). These findings indicate that treatment with LCC-09 alone or in synergism with TMZ effectively inhibit tumorigenicity and suppress GSC phenotypes by significant enhancement of the miR34a/DRD4 ratio.

## 5. Discussion

GBM is a highly infiltrative malignancy and its management is usually plagued by the development of therapeutic resistance and early disease recurrence, despite multi-modal therapy, comprising of surgery, radiation and chemotherapy. This may be attributed to the presence of glioma stem cells (GSCs) which are known to survive radio- and chemotherapy, by circumventing death signals and inducing cell re-population. Recent findings suggest GSCs may be enriched by certain treatment modality [32,33]; these necessitate the development of novel therapeutics capable of targeting GBM cell plasticity and therapy-resistant GSCs. Based on this concept, using a unique approach for discovery of new chemical entities; we developed a novel 5-(2′,4′-difluorophenyl)-salicylanilide derivative, LCC-09. Computer-assisted structure characterization and target prediction suggest LCC-09 targets DRDs and oncogenic markers implicated in GBM (Figure 1). This is particularly important, as the last decade has been characterized by evidence of the complicity of DRD signaling in the anti-GBM therapy resistance, TMZ-associated GSC-expansion, and DRD-induction, dopamine antagonist-elicited suppression of TMZ-related GSC phenotypes, and enhancement of TMZ anticancer efficacy by DRD blockade, as well as the proposal of DRD blockade as effective strategy for the inhibiting the GSC- and therapy-resistant phenotypes in GBM [32,33]. Consistent with the above, we demonstrated that enhanced CD133 positivity and constitutive dopaminergic signaling characterize TMZ resistance in GBM cells (Figure 2). We established a strong positive correlation between the expression of the cell surface glycoprotein CD133 and dopamine receptor DRD4 in CD133+ TMZ-resistant GBM cells (Figure 2). CD133 which is encoded by the prominin-1 (PROM1) gene is well known as a marker of cancer stemness, and more so in GBM, where it has been shown to be essential for GSC maintenance, serving as a marker of cells with enhanced capacity for neurosphere growth initiation and formation of heterogeneous tumors in xenograft glioma mice models [34]. In fact, Liu Q, et al, in their molecular characterization of CD133+ GBM cells, showed that CD133+ GBM cells sorted from neurospheres also expressed CD44 and SOX2, and exhibited enhanced propensity for clonal self-renewal and production of rapidly-growing CD133- progeny, which constitute the major population of cells within the neurosphere niche [35]. Consistent with their finding and ours, we posit that while CD133+ GBM cells may be molecularly quiescent or clinically indolent before they undergo proliferative cell division (PCD) to produce CD133- progeny, it is probable that DRD4 signaling induces the self-renewal of CD133+ GBM cells, enhances PCD which facilitates tumor cell regeneration and disease recurrence in patients with GBM. Having predicted the anti-DRD potential of LCC-09 and established the complicity of DRD4 expression and/or activity in the GSC-phenotype of GBM cells, we provided evidence that LCC-09 suppresses the viability and metastatic phenotype of GBM cells by dysregulating DRD4-mediated AKT/mTOR signaling (Figure 3). Our findings are consistent with the emerging recognition of the integral involvement of the nervous system and the critical role of neurotransmitters as active players in oncogenesis and metastasis, especially as cancerous cells are increasingly being implicated in the transduction of intracellular signaling which are mediated by neurotransmitters and induce the tumorigenesis, tumor growth, and cancer metastasis [36]. Subsequently, we demonstrated that LCC-09, by targeting DRD4 dopaminergic signals and ALDH activity, induces marked attenuation of the stem cell-like phenotype of GBM cells (Figure 4). This may be partially inferred from a recent work suggesting a link between the generation of reactive dopamine metabolites which are autotoxic to dopaminergic neurons and the inhibition of ALDH [37], and is corroborated by the findings of Dolma S et al., that recently highlighted the prognostic relevance of DRD4 expression in patients with GBM based on analysis of The Cancer Genome Atlas (TCGA) data, and suggested that the inhibition of DRD4 can attenuate the proliferation and survival of GSCs by disrupting platelet-derived growth factor receptor β (PDGFRβ)-ERK1/2 and mTOR signaling [38].

Against the background of TMZ being the current chemotherapeutic of choice for patients with GBM and intrinsic or adaptive TMZ-resistance being a major therapeutic sore-spot in the treatment of these patients, it is clinically relevant that LCC-09 synergistically enhances the anticancer effect of TMZ in therapy-resistant GBM cells (Figure 5), and is in line with contemporary therapeutic practice for TMZ-resistant disease, such as combination of TMZ with MGMT inhibitor O^6^-benzylguanine (O^6^-BG), Interferon-β (IFN-β), valproic acid, poly ADP ribose polymerase (PARP) inhibitor, histone deacetylase (HDAC) inhibitor (SAHA), or even oncolytic adenovirus [39]; highlighting the necessity and therapeutic promise of multi-agent therapy over single agent TMZ therapy for effective treatment of GBM. Finally, we showed that LCC-09 suppresses glioma stem cell-related oncogenic and dopaminergic signals by upregulating miR-34a in GBM cells, in vitro and in vivo (Figure 6 and Figure 7). This is consistent with accruing evidence of the broad range of miRNA targets and the ability of miRNAs to impact several components of the hallmarks of cancer, as well as modulate many pathways that are critical for cancer initiation and progression, including proliferation, apoptosis, metastasis, cancer stemness, and drug resistance. Recently, Møller HG, et al. in a systematic review, documented that while a large number of miRNAs are significantly over-expressed in GBM relative to normal brain tissue, 95 miRNAs were significantly underexpressed, including miR-34a [40], with the loss-of-miR-34a-function being associated with high-grade glial neoplasia; in fact the ability of LCC-09 to induce re-expression of miR-34a is of therapeutic significance, considering that miR-34a targets and downregulates several markers of oncogenicity, GSC and TMZ-resistance, such as PDGFRA, c-Met, NOTCH1, NOTCH2, BCL-2, SMAD4, CCND1, SIRT, and CDK6 [41,42], thus, suggesting its role as a tumor suppressor and actionable therapeutic element. This putative tumor suppressor role of miR-34a was also alluded to by Parodi F et al. in their work on the epigenetic landscape and its dysregulation in neuroblastoma, thus further given credence to the validity of our present finding [43]. In the context of miR-34a therapeutic implication, recently, the clinical feasibility of miR-34a as a replacement therapy for patient with cancer began garnering attention, with miR-34a mimics serving as the first miRNA-based therapeutic undergoing a clinical trial [44]. We find it intriguing that TMZ alone had no apparent effect on miR-34a expression, while in combination with LCC-09, there is a 3.61-fold (*p* = 0.005) increase in the expression of miR-34a, though not as much as that elicited by LCC-09 alone; thus we posit that the elicited combinatorial increase in miR-34a is attributable to the pharmacological activity of LCC-09 alone, against the background of common knowledge that despite being first-line treatment drug for GBM, TMZ efficiency is often modest, especially as most cancerous cells are intrinsically resistant or “acquire resistance to TMZ at pharmacotherapeutic concentrations” [45]. While it is also probable that TMZ and LCC-09 compete for same uptake transporters or receptors, with TMZ competitively suppressing LCC-09 ability to induce miR-34a expression, this would be antithetical to our findings and evidence provided, thus, another line of rationalization that is even more probable is competitive potentiation, wherein LCC-09 co-administration with TMZ alters the cyto-availability of TMZ, increasing its intracellular permeation and subsequent induction of the tumor suppressor miR-34a; or better still, the context of pharmacokinetic inverse agonism [45], wherein upon co-administration of LCC-09 and TMZ, TMZ selectively interacts with and binds to the inactive pharmacological conformation of LCC-09, consequently resulting in reduced basal activity of LCC-09 and invariable lesser miR-34a induction. While our understanding of the underlying mechanism for the observed association between TMZ, LCC-09 and miR-34a is inconclusive and continues to evolve, the silver lining remains that combining LCC-09 and TMZ enhances the re-expression of miR-34a significantly, compared to TMZ monotherapy. It is therefore of clinical relevance that we now provide evidence indicating that LCC-09 enhances the expression of this master effector of tumor suppression, as upregulation and/or re-expression of miR-34a potently impedes cancer-promoting phenotypes, including cancer stemness, metastasis, and therapy resistance [44].

## 6. Conclusions

In conclusion, as depicted in our schematic abstract (Figure 8), our results highlight the therapeutic efficacy of LCC-09 as a regulator and/or disruptor of multiple oncogenic signaling and crucial drivers of GBM progression or TMZ resistance, such as DRD4, CDK6, and AKT, through the upregulation or re-expression of master tumor suppressor, miR-34a. The present study lays the foundation for further pre-clinical exploration and clinical utility of LCC-09 alone or as a synergistic enhancer of TMZ anticancer efficacy in GSC—driven treatment-resistant or recurrent GBM cells, with the promise of improving survival rates among patients with GBM.

## Figures and Tables

**Figure 1 cancers-11-01442-f001:**
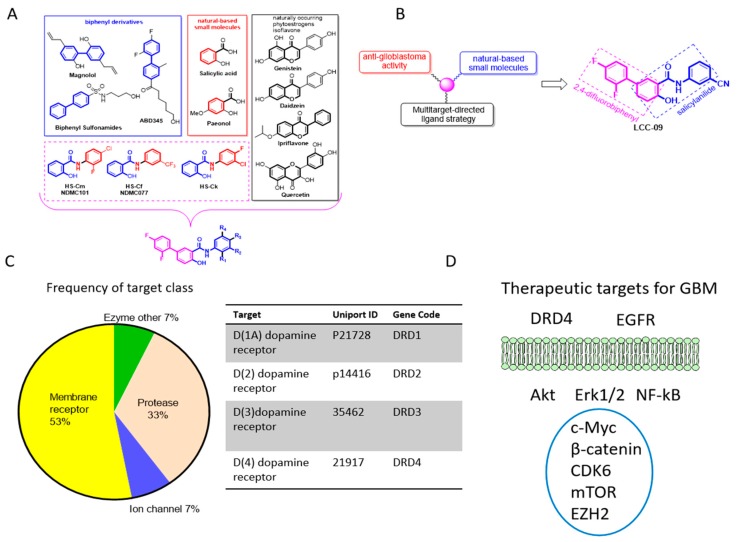
The design, synthesis, and functional characterization of the novel multi-target small molecule dopamine antagonist, LCC-09. Schema showing (**A**) several novel series of 5-(2′,4′-difluorophenyl)-salicylanilide derivatives based on difluorobiphenyl and salicylanilide scaffolds, and (**B**) the pharmacological rationale for the synthesis of LCC-09, consisting of 2,4-difluorophenyl and paeonol moiety of salicylate. (**C**) Pie-chart showing computer-assisted predicted targets of LCC-09 and the target class frequency (**left panel**), and a list of predicted LCC-09-targeted dopamine receptors DRD1, DRD2, DRD3, and DRD4 (**right panel**). (**D**) A schema of LCC-09-relevant therapeutic targets of glioblastoma (GBM).

**Figure 2 cancers-11-01442-f002:**
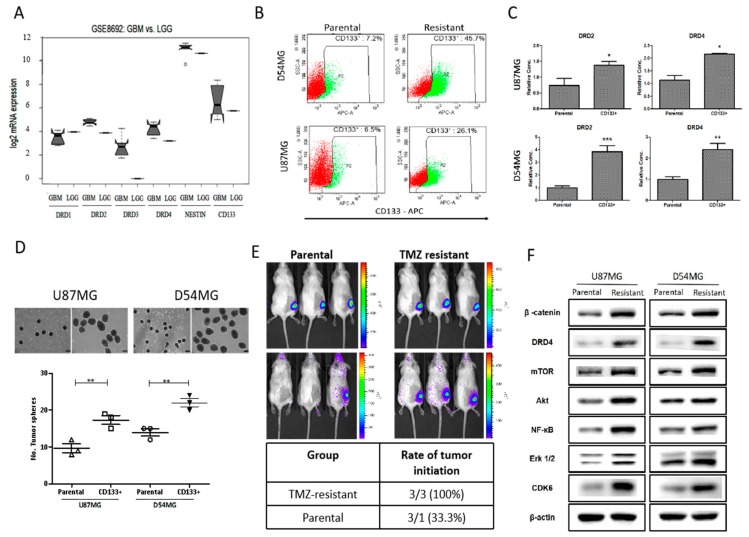
Enhanced CD133 positivity and constitutive DRD4 dopaminergic signaling characterize TMZ resistance in GBM cells. (**A**) Notched box-plot of the differential expression of DRD1–DRD4, Nestin, and CD133 mRNA in low grade glioma and GBM cases from the GSE8692/GPL96 cohort. (**B**) Charts showing the variation in CD133^+^ population in parental or TMZ-resistant U87MG and D54MG cells. (**C**) Graphical representation of the difference in DRD2 and DRD4 between parental or TMZ-resistant CD133^+^ U87MG and D54MG cells. (**D**) Photo-images (upper panel) and histograms (lower panel) depicting the difference in DRD2 and DRD4 between parental or TMZ-resistant CD133^+^ U87MG and D54MG cells. (**E**) Images from day 8 (upper panels) and day 21 (lower panels) bioluminescence analyses showing higher tumor initiation efficiency in mice inoculated with TMZ-resistant CD133^+^ U87MG than the parental group by day 21. (**F**) The expression level of DRD4, β-catenin, Akt, mTOR, NF-κB, Erk1/2, and CDK6 protein are upregulated in TMZ-resistant CD133^+^ U87MG and D54MG cells compared to their parental counterparts. β-actin served as loading control. * *p* < 0.05, ** *p* < 0.01, *** *p* < 0.001; Data represent the means ± SD of experiments performed 4 times in triplicates.

**Figure 3 cancers-11-01442-f003:**
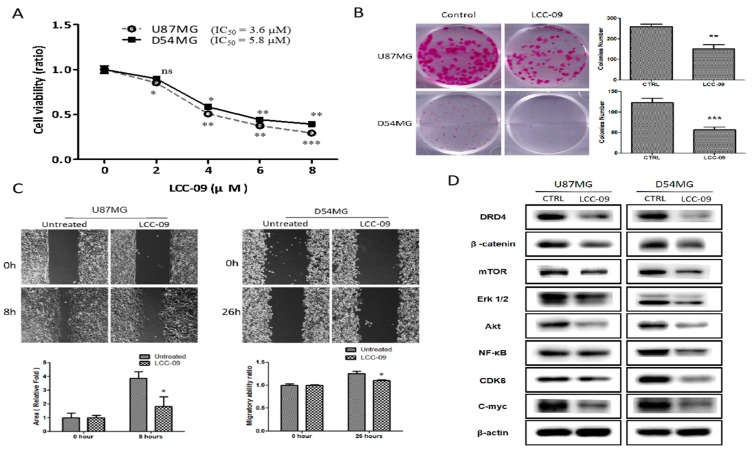
LCC-09 suppressed the viability and metastatic phenotype of GBM cells through the dysregulation of DRD4-mediated AKT/mTOR and NF-κB signaling axes. (**A**) LCC-09 significantly suppressed the viability of TMZ-resistant CD133^+^ U87MG and D54MG cells in a dose-dependent manner. Photo-images and graphical representation of the inhibitory effect of LCC-09 on the ability of TMZ-resistant CD133^+^ U87MG and D54MG cells to (**B**) form colonies, and (**C**) migrate. (**D**) LCC-09 downregulates the expression level of DRD4, β-catenin, Akt, mTOR, NF-κB, Erk1/2, c-Myc, and CDK6 proteins in TMZ-resistant CD133^+^ U87MG and D54MG cells compared to their vehicle-treated counterparts. β-actin served as loading control. * *p* < 0.05, ** *p* < 0.01, *** *p* < 0.001; ns, not significant; Data represent the mean ± SD of experiments performed 4 times in triplicates.

**Figure 4 cancers-11-01442-f004:**
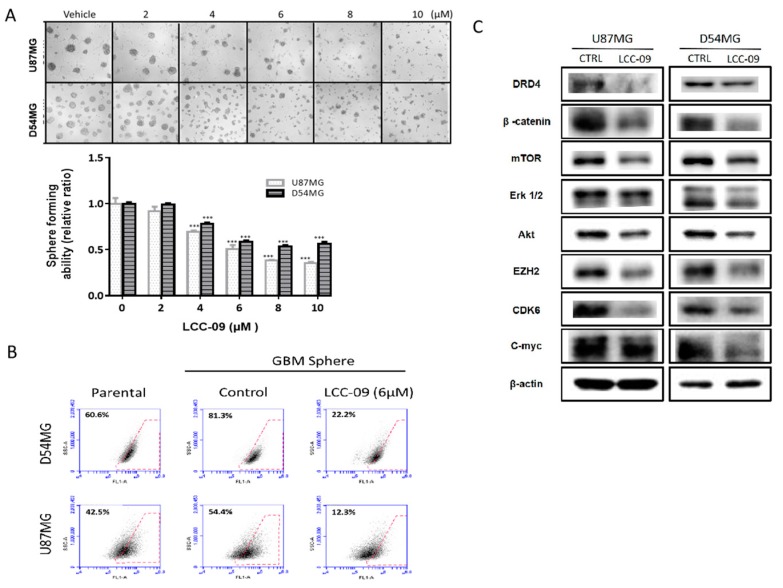
LCC-09 treatment concomitantly suppresses multiple oncogenic and stemness markers as well as aldehyde dehydrogenase (ALDH) activity in GBM cells. LCC-09 (**A**) inhibited the ability of TMZ-resistant CD133^+^ U87MG and D54MG cells to form neurospheres dose-dependently, (**B**) reduced the ALDH activity of TMZ-resistant CD133^+^ U87MG and D54MG cells, and (**C**) suppressed the expression level of DRD4, β-catenin, Akt, mTOR, Erk1/2, EZH2, c-Myc, and CDK6 proteins in the LCC-09-treated TMZ-resistant CD133^+^ U87MG and D54MG cells compared to their vehicle-treated counterparts. β-actin served as loading control. * *p* < 0.05, ** *p* < 0.01, *** *p* < 0.001; data represent the mean ± SD of experiments performed 4 times in triplicates.

**Figure 5 cancers-11-01442-f005:**
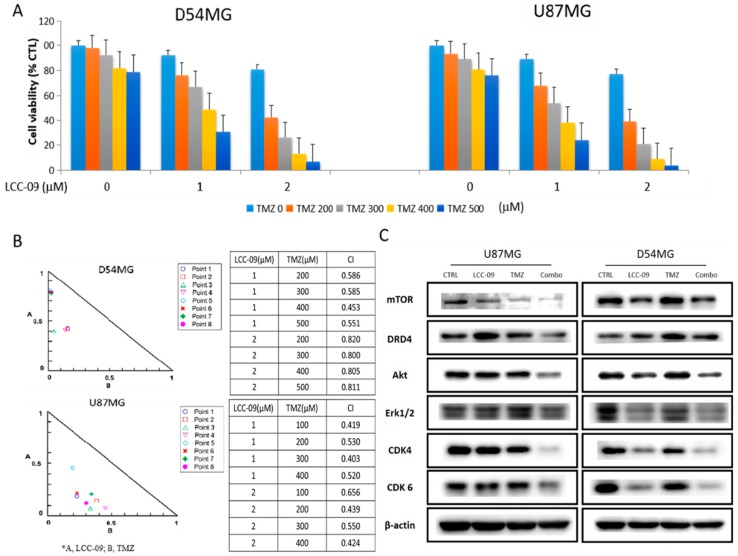
LCC-09 synergistically enhances the anticancer effect of TMZ in therapy-resistant GBM cells. (**A**) Graphical representation of the effect of TMZ alone or in combination with LCC-09 TMZ-resistant CD133^+^ U87MG and D54MG cells. Data represent the mean ± SD of experiments performed 5 times in triplicates. (**B**) Representative right-angle isobologram triangles showing the TMZ/LCC-09 combination points with indicated concentrations and CIs in TMZ-resistant CD133^+^ U87MG or D54MG cells. (**C**) Western blot data showing the effect of LCC-09 and/or TMZ on the expression level of DRD4, Akt, mTOR, Erk1/2, CDK4, and CDK6 proteins in the TMZ-resistant CD133^+^ U87MG and D54MG cells compared to the vehicle-treated control cells. Data represent the mean ± SD of experiments performed 4 times in triplicates. β-actin served as loading control.

**Figure 6 cancers-11-01442-f006:**
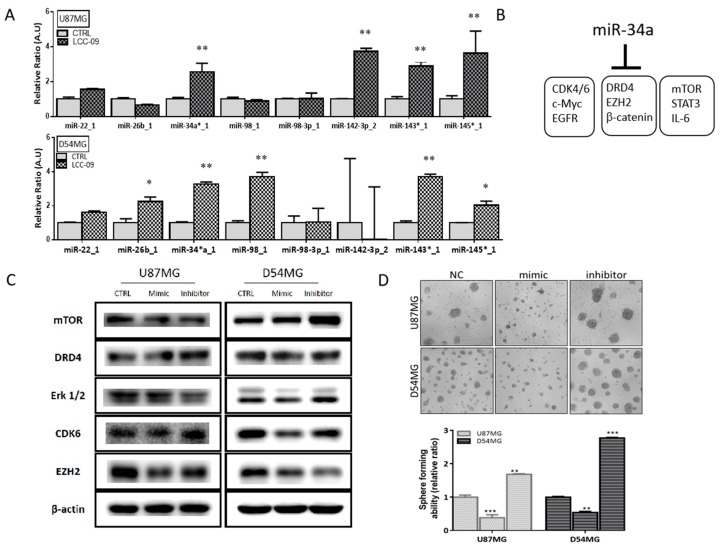
LCC-09 suppresses glioma stem cell (GSC)-related oncogenic and dopaminergic signals while upregulating miR-34a in GBM cells, in vitro (**A**) Graphical representation of the effect of LCC-09 on the expression of miR22, miR-26b, miR-34a, miR-98, miR-98-3p, miR-142-3p, miR-143, and miR-145 in the TMZ-resistant CD133^+^ U87MG and D54MG cells. (**B**) Schema showing LCC-09-relevant targets of miR-34a. (**C**) The differential effect of miR-34a mimic or inhibitor on the expression level of DRD4, mTOR, Erk1/2, EZH2, and CDK6 proteins in the TMZ-resistant CD133^+^ U87MG and D54MG cells compared to the sham siRNA-transfected control cells by Western blot assay. (**D**) Photo-images and histograms of the effect of miR-34a mimic or inhibitor on the neurosphere formation capability of TMZ-resistant CD133^+^ U87MG and D54MG cells compared to the sham siRNA-transfected control cells. β-actin served as loading control. * *p* < 0.05, ** *p* < 0.01, *** *p* < 0.001; data represent the mean ± SD of experiments performed 4 times in triplicates.

**Figure 7 cancers-11-01442-f007:**
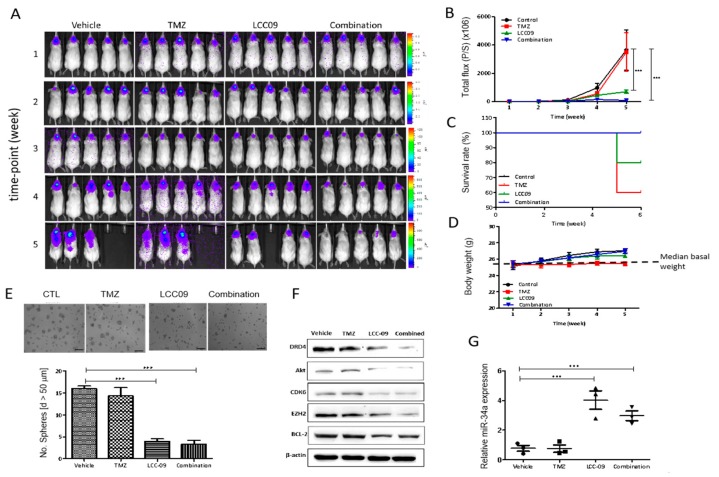
LCC-09 significantly suppresses GSC-related oncogenic and dopaminergic signals by up-regulating miR-34a in GBM cells, in vivo. (**A**) Bioluminescence imaging of the effect of LCC-09 on tumor burden from week 1 to week 5, and (**B**) semi-quantitative analysis of bioluminescence total flux (photons/s) of the tumor burden in mice inoculated with luciferase-expressing TMZ-resistant CD133^+^ U87MG cells, in comparison to treatment with vehicle. The differential effect of LCC-09 and/or TMZ, on (**C**) the survival, using the Kaplan–Meier plot, and (**D**) body weight of treated mice, compared to those in the control group. (**E**) Photo-images of the effect of LCC-09 and/or TMZ on the neurosphere-forming capability of the cells from tumor samples harvested from the treated mice, compared to the control group. (**F**) The effect of LCC-09 on the expression of DRD4, ERK1/2, EZH2, and CDK6 in tissue samples from mice inoculated with luciferase-expressing TMZ-resistant CD133^+^ U87MG cells, in comparison to treatment with vehicle. (**G**) Graphical representation of the relative miR-34a expression level in mice inoculated with luciferase-tagged TMZ-resistant CD133^+^ U87MG cells and treated with LCC-09 or vehicle. P/S, photon/s; * *p* < 0.05, ** *p* < 0.01, *** *p* < 0.001; data represent the mean ± SD of experiments performed at 4 times in triplicates.

**Figure 8 cancers-11-01442-f008:**
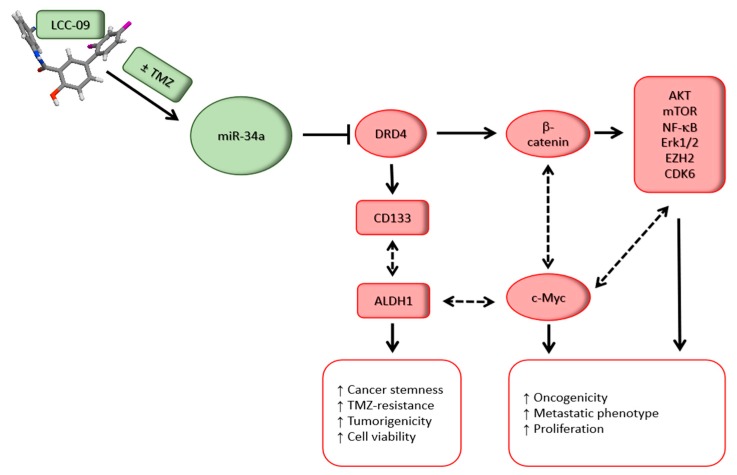
Schematic abstract depicting how the novel multi-targeted small molecule, LCC-09 inhibits the stemness, oncogenic and therapy-resistant phenotypes of glioblastoma cells by increasing miR-34a and deregulating the DRD4/Akt/mTOR signaling axis.

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
