# Peer review of "A Novel Multi-Target Small Molecule, LCC-09, Inhibits Stemness and Therapy-Resistant Phenotypes of Glioblastoma Cells by Increasing miR-34a and Deregulating the DRD4/Akt/mTOR Signaling Axis"

_cancers, 2019, doi:10.3390/cancers11101442_

Round 1

Reviewer 1 Report

I. In this study, the authors test the anti GBM effect of a small molecule LCC-09 in vitro and in vivo and try to characterize the mechanism of the anti-tumor function.

LCC-09 is a multi-target molecule.

2.LCC-09 mediate reduction of viability, clonogenicity and migaration in TMZ-resistant GBM cells in vitro.

LCC-09 suppresses DRD4 expression and the expression of key components of AKT/mTOR and NF-kb signaling axes. LCC-09 inhibits TMZ-resistant GBM cells' stemness, inhibits ALDH activity. In ALDH hi population, stem-like phenotype related genes also have been downregulated. LCC-09 works synergistically with TMZ. LCC-09 upregulates miR-34a, and miR-34a mimic can do the similar thing as LCC-09 to the GBM cells. miR-34a inhibitor does in the opposite way. LCC-09 anti-tumor effect and the synergistical effect with TMZ can be replicate in animal model.

II.The study fits the journal with certain novelty and impact.

questions related to points above:

1.The targets of LCC-09 was predicted by computer algorithm. Is there any direct evidence that this molecule can bind or interfere with any target? It seems this molecule interact with a whole lot of targets like DRD1/2/3/4 and EGFR, Akt, mTOR, Erk1/2, ... CDK6, and EZH2.

3.If LCC-09 can target so many molecules, it's hard to figure out the actual mechanism. We observed the deregulation of DRD4 and all other components from AKT/mTOR and NF-kb signaling axes. Is it a pan- effect from LCC-09 or the suppression of DRD4 causes the deregulation of the pathway?

4.The same question: ALDH inhibitor is the cause of loss stemness or just another concomittant phenomenon?

6.Finally, the author seems to attribute the function of LCC-09 to the upregulation of miR-34a, but there seems to be missed links. How's the interaction between LCC-09 and miR-34a?

This study described a novel molecule and its potential function and mechanism. Definitely, more studies are needed to elucidate the mechanism, but I don't think that should be included in this paper.

III. Other comments:

Major one: as authors' description, figure 3D is the western result from cells treated by 3uM LCC-09; figure 4C is the western result from cells treated by 6 uM LCC-09 and only for the ALDHhi population. They should be two separate experiments, but I found the western photos were actually the same except the EZH2 bands. There must be some mistake which should be fixed.

Minor ones:

1.Please indicate the time points of animal images in Figure 2E and Figure 7A.

2.Fix the figure legend alphabetic (F) and (G) in line 466 and line 469.

Reviewer 2 Report

Recommend a summary graphic/figure demonstrating the relationship, mechanism of action, target sites of all the various factors implicated in this manuscript.

Reviewer 3 Report

In this manuscript Wen and Coll. describe the cellular and molecular effects of the small molecule LCC-09, by using GBM in vitro and in vivo models.

Most specifically, Authors convincingly describe the use of LCC-09 as anti-tumor molecule, directed against GSC population. The appropriate use of LCC-09 against glioma cells is demonstrated through several cellular and approaches, both alone and in combination with temozolomide.

As a whole, the manuscript reports convincing data even if some major issues are to be solved:

1) Materials and methods are lacking of an entire section referred tomicroRNA expression and real-time PCR analysis, further to transfection experiments. Authors are encouraged to add these sections

2) The following sentence:" The combination of indicated concentrations of TMZ with 157 LCC-09 in fixed ratio" (lanes 157-158) is incomplete

3) Authors should specify how many cells were startly seeded in the scratch wound-healing migration assay and if the cells were cultured w/o serum after the scratch and how they quantified the closure of the healing

4) In the results section the titles of each paragraph must be in bold

5) In figure 1C's legend Venn diagram should be replaced with Pie-chart

6) In figure 2A whiskers are lacking

7) The expression of pERK should be evaluated in addition to that of total ERK. For all WB, it is not clear if the represented plot is only representative of more experiments (in that case, an histogram showing mean and std dev values for each protein assayed is higly appreciated) or if the experiment has been performed as single copy (in that case Authors should specify this).

8) In the legends where a statistic is indicated, Authors should specify the type of statistical test used

9) In figure 3A add std dev and statistical significance

10) Lane 403 6 mM should became 6 uM

11) Lane 407, Authors should cite the following reference: PMID: 27751904

12) Figure 7G reports an increase of miR-34a expression that in the condition (LCC-09 + TMZ) is less marked than LCC-09-only treatrment. This is in contrast with the data reported in the remaining panels of figure 7. Authors should explain this discrepancy

13) In figure 7C the type of survival test used must be specified

14) English should be revised throughout the text (e.g.: lane 529 miRNAs targets should become miRNA targets) 

Round 2

Reviewer 1 Report

I am quite satisfied with their response. I recommend to accept this manuscript for publication.

Author Response

We thank the reviewer for taking time to go through our revised manuscript and agreeing to accept our work for publication. We are most grateful.

Reviewer 3 Report

In this revised version, Authors improved several aspects of the manuscript but several issues must be fixed yet, before considering the manuscript for publication:

1) A7: We thank the reviewer for pointing this out. We have now corrected this in the revised manuscript. Please kindly see our revised Results section, Page 9, Lines 347-359:

Whiskers are still missing in the updated figure 2A

2) A8: We appreciate the reviewer’s comment. All assays in this present study were performed at least 4 times in triplicates. Please kindly see our Materials & Methods section, Statistical Analysis section, Page 6, Lines 274-279:

p-ERK data are still missing in the updated version of the paper

3) A9: We thank the reviewer for this comment. We are not sure we fully understand what the reviewer means here and humbly apologize for this. However, may we politely refer the reviewer to our Materials & Methods section, Statistical Analysis section, Page 6, Lines 274-279:

Authors should specify the type of statistical test used for each experiment and report it in the figure legend

4) R10: We thank the reviewer for this comment. Actually the figure in question does include standard deviation as indicated by 2-sided error bars, however these are almost not apparent because of the closeness of the experimental values to the mean/average value. We have indicated no statistical significance since there was no comparison between the U87MG and 54MG cells’ viability.

Authors should calculate significance based on the comparison between treated and control cells and report these data in the graph

5) A12: We thank the reviewer for this suggestion. While we would have loved to cite all theme-relevant works in acknowledgement of authors’ contribution to the field, we however politely request to pass on this one considering that (i) we already have a long list of references for an original research paper, and (ii) the suggested paper is addresses neuroblastoma, not GBM.

Although referred to NB, the reference cite miR-34, corroborating its potential role as epigenetic regulator also in the disease model you are studying. This is why this Reviewer suggested to add this reference to the manuscript  

6) A13: We thank the reviewer for this observation. While we would have wished that this particular data was as described by the reviewer and as we hypothesized, however, as with all our work, we have presented our results as obtained without letting our intent or bias affect data presentation. The silver lining though is that combining LCC-09 and TMZ enhanced the re-expression of miR-34a significantly, compared to TMZ monotherapy. Please kindly see our revised Results section, Page 14, Lines 461-493:

About this aspect, it is true that miR-34a shows a less increase in the condition “combination” with respect to LCC-09 condition, as compared to NC and TMZ samples, but this datum doesn’t explain why its targets decrease in the same condition. Authors should discuss this discrepancy, eventually giving a less importance to miR-34a, and avoiding to cite it in the title  

Round 3

Reviewer 3 Report

Authors improved their manuscript.

Last annotation is about the presentation of data on p-ERK. Both p-ERK and total ERK should be shown and quantified (in association with the housekeeping protein) in a same blot. Authors are encouraged to do that and to comment their result, accordingly.

Author Response

Q1: Reviewer #3: Authors improved their manuscript.

A1: We sincerely thank the reviewer for the time taken to review our work once again, clarifying some of the initial concerns and giving us the opportunity to address those concern appropriately. As alluded earlier, we continue to believe the reviewer has our interest at heart and this will help improve the quality of our paper even more.

Q2: Reviewer #3: Last annotation is about the presentation of data on p-ERK. Both p-ERK and total ERK should be shown and quantified (in association with the housekeeping protein) in a same blot. Authors are encouraged to do that and to comment their result, accordingly.

A2: We appreciate the reviewer’s comment. As requested by the reviewer, we have now presented the data such that both p-ERK and total ERK are quantified and shown (in association with the housekeeping protein) in same blots.” Please kindly see our updated Supplementary Figure S3 and its legend:

Suppl. Figure S3. LCC-09 alone or in combination with TMZ significantly suppress activation of Erk1/2. Western blot images of the effect of LCC-09 on the expression of p-Erk1/2 and Erk1/2 proteins in (A) parental U87MG and D54MG cells or (B) neurospheres derived from U87MG and D54MG cells. (C) The effect of LCC-09 and/or TMZ on the protein expression levels of p-Erk1/2 and Erk1/2 in TMZ-resistant U87MG or D54MG cells. b-actin served as loading control. Inscribed numbers, relative protein expression based on densitometry using the NIH ImageJ software (https://imagej.nih.gov/ij/).
